# Use of Biopolymers in Mucosally-Administered Vaccinations for Respiratory Disease

**DOI:** 10.3390/ma12152445

**Published:** 2019-07-31

**Authors:** Margaret R. Dedloff, Callie S. Effler, Alina Maria Holban, Monica C. Gestal

**Affiliations:** 1Department of Biology, Clarkson University, Potsdam, NY 13699, USA; 2Department of Natural Sciences and Mathematics, College of Arts and Sciences, Lee University, Cleveland, TN 37311, USA; 3Department of Microbiology, Faculty of Biology, University of Bucharest, 030018 Bucuresti, Romania; 4Research Institute of the University of Bucharest (ICUB), 050107 Bucharest, Romania; 5Department of Science and Engineering of Oxide Materials and Nanomaterials, Faculty of Applied Chemistry and Materials Science, University Politehnica of Bucharest, 1–7 Polizu Street, 011061 Bucharest, Romania; 6Department of Infectious Diseases, College of Veterinary Medicine, University of Georgia, Athens, GA 30602, USA

**Keywords:** polymeric vaccines, respiratory infections, mucosal administration, immunomodulation, nanovaccines

## Abstract

Communicable respiratory infections are the cause of a significant number of infectious diseases. The introduction of vaccinations has greatly improved this situation. Moreover, adjuvants have allowed for vaccines to be more effective with fewer adverse side effects. However, there is still space for improvement because while the more common injected formulations induce a systematic immunity, they do not confer the mucosal immunity needed for more thorough prevention of the spread of respiratory disease. Intranasal formulations provide systemic and mucosal immune protection, but they have the potential for more serious side effects and a less robust immune response. This review looks at seven different adjuvants—chitosan, starch, alginate, gellan, β-glucan, emulsan and hyaluronic acid—and their prospective ability to improve intranasal vaccines as adjuvants and antigen delivery systems.

## 1. Introduction

Respiratory diseases pose a significant threat to the length and quality of life. While a large proportion of respiratory conditions are chronic disorders, such as asthma and chronic obstructive pulmonary disease (COPD), communicable respiratory diseases are a major threat to global health [1]. Pneumonia, tuberculosis, and influenza are substantial contributors to the burden of disease, and acute lower respiratory infections alone caused approximately 2,736,000 deaths in 2015 [2]. Even vaccine preventable diseases, such as pertussis, continue to contribute to the global numbers of cases, particularly in young children, with 24.1 million cases in 2014. In developing regions, over 50 percent of childhood deaths can be due to pertussis [3]. Furthermore, influenza, while often thought to be a rather benign disease, causes approximately 25,000 deaths annually in the United States [4]. The currently available vaccines for these diseases have many limitations, making it critical to develop more effective immunizations using better adjuvants [5,6,7,8]. 

While the introduction of vaccination has greatly decreased the incidents of communicable respiratory diseases, general concerns have emerged. Due to the safety issues that arise using live attenuated vaccines, which provide the strongest immune response, many vaccines opt for other routes, such as utilizing dead pathogens, proteins belonging to the microbe, or toxins produced by the pathogen [9]. However, these alternatives do not induce as robust an immune response as the live attenuated vaccines do. To overcome these limitations, adjuvants are currently administered as a component of the vaccine to boost the immune response [9].

For some respiratory diseases, such as pertussis and influenza, it is important to consider building both a robust systemic immunity (protective IgG levels) and mucosal immunity (IgA) to prevent not only symptoms of infection, but also further transmission [10]. While most vaccines currently in use are injectable and induce a systemic immunity, these do a poor job of developing mucosal protection. Intranasal administration of a live attenuated pathogen stimulates both systemic and mucosal responses, leading to a profound and robust protective immunity, but it has the potential for undesirable adverse effects, particularly in younger children and weakened individuals, leading to safety concerns [11]. Additionally, intranasal formulations have retention issues because, unlike injectable vaccines, they have the potential to leak out of the nose and fail to cause their intended effect [12,13,14]. Despite possible side effects, intranasal delivery of vaccines is an attractive route due to the high permeability of the respiratory region of the nose. 

This area of the nasal cavity has a large surface area for adsorption and very vascularized epithelia [15,16]. The expansive surface area is due to the presence of several hundred microvilli per cell, which allows for rapid adsorption and high bioavailability of drugs or vaccines that are delivered intranasally [15]. In addition, the high vascularity of this region of the nasal cavity allows for a strong interaction between the immune system and intranasally administered vaccines facilitating the generation of mucosal immunity. Further research and advancements in the development of safer intranasal formulations for respiratory diseases have the potential to lead to vaccines that are more effective than those currently in use.

The emerging field of nanotechnology is presenting a great opportunity to improve intranasal vaccines for respiratory diseases. Particularly, biopolymers have been used as adjuvants in conjunction with disease-specific antigens in intranasal, oral and subcutaneous administrative routes [10,17]. Coadministration of certain biopolymers with antigens has been shown to correlate with elevated levels of virus/bacteria-specific antibodies, demonstrating an augmented immune response [10,17]. This has the potential to allow for safer formulation strategies, such as using dead pathogens or isolated antigens, to trigger more powerful and robust immunity, as a live attenuated vaccine would. Biopolymers have the best biocompatibility and degradability rates among all biomedical materials, but they can be engineered to ensure the protection of various bioactive molecules, such as antigens and drugs against degradation and defective biointeraction [18]. Biopolymers are also used as drug delivery systems, and as mucoadhesives to aid in the issue of poor retention of medications and antigens in the nasal cavity [9,12,13,14]. The most important particularities and advantages of biopolymers for the design of vaccines are included in Table 1. 

This review compiles literature on recent research surrounding the use of biopolymers as adjuvants, mucoadhesives, and antigen delivery systems, as well as their potential to improve intranasal formulations for respiratory diseases.

## 2. Chitosan

Chitosan is a derivative of chitin that is created by deacetylating chitin to 50% acetylation in the presence of a base. It is a semi-crystalline, cationic polysaccharide made of D-glucosamine and N-acetyl-D-glucosamine [10]. Chitosan is available in many forms depending on the degree of acetylation, which can alter its properties, including its immunological activity. Its versatility, low cost and minimal toxicity make it attractive for use in medicine [26]. 

Due to its capability to bind to negatively charged surfaces, like mucous, chitosan has been shown to be strongly mucoadhesive, resulting in an increase in nasal adsorption [27,28]. This property is advantageous for its use in mucosal vaccines because it facilitates the contact between the antigen the nasal passages by ensuring that the vaccine does not leak out. Additionally, chitosan opens the tight junctions between epithelial cells, which allows for more efficient absorption [24,29]. This effect has been previously observed in coadministration of insulin and chitosan, which resulted in an increase in intranasal absorption, making it a promising adjuvant [10,26,27]. The same has been demonstrated for morphine and chitosan [27]. Importantly, chitosan is a natural component of fungal cell walls, which triggers the activation of the immune system by stimulating cytokine production [17,30]. This property especially makes it appealing for use as an adjuvant. 

Chitosan nanoparticles have been commonly used for vaccine delivery. Once these nanoparticles are combined with antigens, they have been shown to induce an immune response when administered both orally and intranasally [31]. Low molecular weight chitosan nanoparticles are able to generate strong mucosal and humoral immune response following intranasal delivery. The mechanism of action of this delivery system is thought to be different than that of the polysaccharide and antigens acting alone in solution [32]. The cells in the nasal cavity interact with the nanoparticles and allow them to cross the nasal epithelium, and then deliver them to immunocompetent cells, producing a robust immune response that results in the development of memory cells specific for the pathogen of choice [32]. 

When used as an adjuvant, different formulations of chitosan have been shown to elicit a wide variety of immune responses. For example, a study performed by Nishimura et al., showed that some derivatives were able to cause the generation of helper T-cells in mice while other formulations led to the creation of cytotoxic T-cells [33]. There are some chitosan formulations that have been shown to help boost the non-specific, innate response to infection by stimulating the production of interleukins and interferon and activating macrophages in mice [34]. These properties make chitosan an attractive adjuvant because different pathogens require different immune responses, so being able to alter chitosan to induce the desired response is appealing. For example, a Th1 response is important in combating *Bordetella pertussis* infection, but the current acellular vaccine induces a Th2 response, as revealed in mice models and human infants [35,36]. Adjuvants such as chitosan that can be altered to induce different responses would be beneficial for their applications in several vaccines like *B. pertussis*.

Chitosan has been shown to increase the efficacy of intranasally administered vaccines in mice [10]. It has been demonstrated that mice intranasally challenged with two *Bordetella pertussis* antigens (filamentous hemagglutinin—FHA; and pertussis toxin—rPT) in combination with chitosan are able to generate robust systemic and mucosal immunity. Protection in this study was measured as IgG titers present in the serum, and levels of IgA in secretions obtained from the lungs and nasal cavity. The mice vaccinated with the antigens in combination with chitosan revealed higher antibody titers than those treated with antigens alone. Additionally, the mice treated with the antigens via injection presented increased levels of antibodies in the serum, but not in the mucosal membranes which is the most relevant considering that protection against respiratory pathogens is mostly mucosal [10]. This study demonstrated the ability of chitosan to build a robust mucosal response for antigens of choice, indicating that it shows promise as an adjuvant in intranasally administered vaccines.

In a study performed by Illum et al., mice were vaccinated using four different treatments: (1) intranasally challenged with hemagglutinin (HA) and neuraminidase (NA) without chitosan; (2) intranasally challenged with HA and NA with chitosan; (3) subcutaneously challenged with HA and NA without chitosan; and (4) subcutaneously challenged with HA and NA with chitosan. Once again, the subcutaneous vaccination resulted in a serum response. The group that was intranasally vaccinated with HA, NA and chitosan revealed higher immune response in both the serum and respiratory mucosal membranes [18]. Furthermore, it has been shown that an intranasal flu vaccine containing chitosan caused an increase in anti-hemagglutinin IgA in the lungs of vaccinated mice [37]. When used as an adjuvant, chitosan has been able to strongly induce IgA production in mucosal surfaces. High IgA antibody titers in said surfaces are critical for combating future infection with respiratory diseases. 

Overall, chitosan is highly promising as an adjuvant. This polysaccharide has been able to successfully enhance humoral and mucosal immunity as compared with the currently available vaccines, and has shown especially encouraging results in its ability to induce an IgA response in the lungs when delivered intranasally. The most relevant particularities of chitosan in the design of mucosally administered vaccines are presented in Figure 1. The most important properties of chitosan refer to the fact that it can be administered via various routes, depending on the pharmaceutical formulation and biomedical purpose. This polymer was tested as safe and efficient for the main mucosal routes (respiratory, oral, rectal and intravaginal) [38]. Importantly, it can be conjugated with various types of antigens (proteins, LPs, and others) enhancing its interactions with cellular, humoral and mucosal immune response, and subsequently making it more suitable as a vaccine adjuvant [38,39]. Its ability to be loaded with a diverse range of ligands and the tunable biomedical applications rely mainly on the development of improvement programs of chitosan, such as chemical modifications (i.e., hydrophilic modification, hydrophobic modification), specific ligand modification (i.e., galactose, mannose, peptide), M-cell targeting modification, and DNA vaccine [40].

## 3. Starch

Starch is a widely available biopolymer that can be found in everyday materials such as grains and beans. It is non-toxic and has no known interactions with currently available drugs, making it a very commonly used substance for drug delivery [18]. Starch microspheres are very stable and exhibit a slow-release effect [41]. Although older studies determined that starch alone has low immunogenicity, starch derivatives are potential vaccine adjuvants for intranasal immunizations due to their ability to activate the immune response. Furthermore, in recent studies it has been demonstrated that raw starch microparticles have immunostimulant activity in mice vaccinated with BCG and challenged with *Mycobacterium tuberculosis* [42].

Polyacryl starch microparticles have shown promise as adjuvants in mucosally administered vaccines. These microparticles are prepared by reacting acrylic acid glycidyl ester with starch, followed by polymerization in a water-in-oil emersion [43]. The microparticles, conjugated with human serum albumin, have been shown to induce a robust antibody response in mice when administered intraperitoneally, intramuscularly and intravenously [44]. Starch microparticles can be internalized by mucosal tissues and the internalization process is influenced by the conjugated antigen. For example, recombinant cholera toxin B subunit (rCTB), which is known to bind to the ubiquitous GM1-receptor, proved to facilitate the uptake of the conjugated starch microparticles in mice intestinal villi via the GM1-receptor [45]. A study performed by Wilkingsson and Sjoholm revealed that using polyacryl starch microparticles as an adjuvant for a mucosally administered vaccine induced robust mucosal and systemic immune responses in mice [46]. While this study was performed using oral vaccines, it indicates that polyacryl starch microparticles may be viable for use in an intranasal vaccination for respiratory disease.

A study used raw starch microparticles as a carrier system for a *Mycobacterium tuberculosis* antigen in order to examine the ability of the conjugated system to induce antibody and interferon gamma (IFN-y) production, which is critical for the immune response against tuberculosis. The results demonstrated that either intranasally or orally, both methods induced high IgG titers and robust production of IFN-y, revealing that raw starch microparticles are promising for their use in conjugate vaccines [42]. Conjugate vaccines are commonly used in immunizations for respiratory diseases, so starch may be a useful adjuvant for future vaccine development. 

Studies have demonstrated that starch microparticles are able to induce robust immune responses for respiratory pathogens when delivered mucosally. Due to their low cost, minimal toxicity and ability to activate the immune response, these microparticles may be attractive for use as adjuvants in future vaccines.

## 4. Alginate

Alginate is a naturally occurring polymer that usually comes from brown seaweed. It is non-toxic and inexpensive, making it appealing for use in biomedical applications. Alginate has high biocompatibility and low immunogenicity [22]. It possesses mucoadhesive properties, and when used as a coating, decreases the degradation of the antigen. Additionally, alginate has been known to enhance permeability in the nasal cavity [47]. It has previously been used in many applications such as wound dressing and drug delivery, but excitingly, its properties suggest that it could be used as an adjuvant or antigen-delivery system for mucosal vaccines. 

Previous studies have shown that alginate microparticles, when administered orally, elicit both a systemic and mucosal immune response [48]. Additionally, alginate has shown promise as an adjuvant due to its ability to induce the production of cytokines such as TNF-a [49,50]. When pig serum albumin (PSA) with alginate microspheres was intranasally administered in mice, this conjugate induced high IgG1 titers in both serum and nasal secretions. While other antibodies were produced in response to the PSA conjugated to alginate microspheres, there was no significant difference from the control groups [51]. 

Alginate can be combined with chitosan nanoparticles for its use in vaccines and drug delivery. Due to alginate’s anionic properties, it binds well with chitosan, creating dense but stable compounds [52]. Additionally, alginate has been used as a coating for chitosan-based vaccines because it protects the antigen from enzymatic degradation. This is especially useful in the case of intranasal vaccines because of the enzymes present in mucous [53]. There have been several studies that have shown that the combination of alginate and chitosan, when delivered mucosally, have the ability to induce robust mucosal and systemic immune responses [53,54]. Due to its mucoadhesive properties, low cost, and its ability to be combined with adjuvants like chitosan, alginate has an incredible potential for its use in intranasal vaccines for respiratory diseases.

## 5. Gellan

Gellan gum is an acidic polysaccharide made of repeating tetrasaccharide units, produced by the bacterium *Pseudomonas elodeu* [12,13,14]. It is commonly used in food production, as an emulsifier to thicken products and make them more homogenized, as well as an industrial lubricant [14]. The naturally-produced form of gellan is high-acyl (HA) gellan, can be chemically deacetylated to low-acyl (LA) gellan. The ratio of these forms can be changed to obtain different viscosities, which allows for it to be used for different purposes, such as intranasal spraying for gellan of low viscosity [14].

The effect of gellan on increasing the efficacy of a mucosally-delivered vaccine is due to its viscoelasticity [12,14]. When gellan gum encounters cations in the nasal cavity, it interacts with them in the mucous and neutralizes and condenses them via a cross-linking mechanism, forming helices and becoming an adherent gel [12,13,14]. This property is highly useful because it allows for gellan to be in a less viscous state that can be administered as an intranasal spray before condensing and becoming more viscous and adherent on contact with the mucous. One of the major limitations with intranasal delivery is that the antigens are likely to not reach their intended targets because the vaccine tends to run back out of the nose. Gellan adheres to the mucosal surface of the nose, allowing for an increase in antigen retention and the consequent generation of an immune response [12,13,14]. 

As a result of its viscoelasticity and adherent characteristics, gellan enhances the mucosal immune response to influenza when used as an intranasal vaccine adjuvant [17]. Gellan also promotes a systemic response when administered intranasally in mice, evidenced by an increase in IgG levels in the serum. Similarly, the mucosal response is enhanced, which is evidenced by an increase in IgA titers in fluids from the nasal cavity and lungs [13,17]. Furthermore, in this work, mice revealed higher IgG titers in the serum for influenza A than for influenza B, suggesting that gellan particularly promotes immunity against influenza A [13]. 

Gellan is a promising candidate for an intranasal vaccine adjuvant because, like chitosan, it acts as a mucoadhesive agent. While not as powerful as chitosan, it too has been shown to increase serum and localized mucosal immunity when used as an adjuvant in intranasally administered vaccines. Importantly, while chitosan activates the immune system and has mucoadhesive properties, gellan has only mucoadhesive properties [17]. The contradictory levels of efficacy between coadministration of gellan with influenza A and B antigens reveals that more research is needed, with more various antigens, to better characterize how gellan could be used effectively on a larger scale [12,17].

## 6. β-Glucans

β-glucans are polysaccharides that are natural components of fungal cells. Specifically, they are components of glucan particles, emptied cell walls of yeast cells, which can be used as a delivery system for vaccines [55,56]. β-glucan and glucan particles are fungal specific, causing more evolved organism to recognize them as a threat and activate the innate immune system, and consequently stimulate an inflammatory response [57]. β-Glucans also activate complement through interaction with the complement receptor 3. Interactions with the immune response happen via the dectin-1 pathway, in which the β-glucan receptor works with toll-like receptors to upregulate cytokines IL-4 and IL-13, activating macrophages, and leading to the proliferation of CD4 and CD8 cells [58]. These polysaccharides can act as an immunostimulants by enhancing natural killer cell function [59]. 

β-glucans are also often linked to chitin, whose derivative, chitosan, has immunomodulatory properties promoting its use in research focused on mucosally administered vaccines. Conjugation with chitin does not alter the interactions with the host response and it still stimulates the immune response in the same way as β-glucans [60]. Mice studies using ovalbumin embedded inside of glucan particles showed that this method of administration led to increased levels of CD8 and CD4 cells in the bone marrow compared to treatment with ovalbumin alone [61].

A recent study examining the ability of β-glucan particles to deliver oral vaccines has indicated that these particles are successful at enhancing immune response to the antigens. Using an oral vaccine with β-glucan skewers towards a Th-17 response, increasing the production of IgA antibodies, suggesting that using β-glucan particles in vaccines may be a potential option to prevent respiratory infections [62]. 

Overall, β-glucan particles show promise as an intranasal vaccine adjuvant for respiratory diseases due to their stimulation of the dectin-1 pathway, as well as for use in glucan particles as a delivery system for oral vaccines. The possibility of combining β-glucan and chitosan is particularly promising for further research and advancement.

## 7. Emulsan

Emulsan is an exopolysaccharide produced by RAG-1, a strain of *Acinetobacter calcoaceticus*. *A Acinetobacter calcoaceticus* metabolizes hydrocarbons, producing an acetylated polymer made of repetitions of three amino sugars with variable fatty acid side chains, with varying degrees of substitution [63,64]. These molecules are amphipathic, allowing them to form emulsions between oil and water, similar to gellan [64], and they are also similar to lipopolysaccharides in their high level of variability leading to different types of immune responses [65,66]. Emulsan was shown to stimulate macrophages to release TNF-α in vivo, leading to inflammation [63]. Deacetylating the emulsan particles no longer caused macrophage activation, suggesting that the fatty acids may be required for the activating effect and not the sugars [66]. However, modifying the fatty acid side chains can influence the strength of activation. The mechanisms by which emulsan leads to a macrophage activation is by using the TLR4 receptor, these induce an increase in IgG2a [66,67]. 

In several cases, emulsan has been proven to be an equally or more effective adjuvant than alum for injected vaccines, and no toxicity has been observed in injectable vaccinations containing emulsan [67]. It has been demonstrated that one cerulenin, a particular emulsan, successfully increases the immune response when administered intranasally in mice [67]. Similar to glucan particles and alginate, emulsans can also be used as a vaccine delivery system, forming microspheres and carrying antigens and other adjuvants inside of their hydrophobic cores [9], suggesting that emulsans could be used to further improve injectable and intranasal vaccines. Emulsans show potential as adjuvants due to their tailorability, low toxicity, and immunomodulatory properties. 

## 8. Hyaluronic Acid

Hyaluronic acid is a polymer, composed of two sugars, namely, N-acetyl-glucosamine-6-phosphate and glucuronic acid. This is a bioadhesive polymer, which has been investigated for numerous biomedical applications, including vaccines. Along with their mucoadhesive properties, the utilization of hyaluronic acid polymers provides a powerful approach for improving the immunogenicity of a wide variety of antigens [68,69]. As a vaccine enhancer, hyaluronic acid (HA) has been investigated in combination with various particles, such as liposomes. In order to obtain efficient polymeric structures for vaccine design, thiolated HA was used by conjugation of HA with L-cysteine via EDC/NHS reaction. Such polymeric structure showed improved colloidal stability and prolonged antigen release, inducing potent humoral immune responses with increased IgG titers in sera in mice [70]. Cationic liposome-hyaluronic acid hybrid nanoparticles act as efficient delivery vehicles for protein antigens and immunostimulatory agents in vitro and in vivo, while being well suited for intranasal vaccination in a mouse model [70]. 

These hybrid nanoparticle systems have been known to induce dendritic cell (DC) maturation by first determining the upregulation of costimulatory molecules, including CD40, CD86, and MHC class II, which greatly contribute to an enhanced specific T cell and antibody response following intranasal vaccination [71]. The use of hyaluronic acid polymers for mucosal delivery of vaccine antigens and adjuvants has been patented since 1999 (https://patents.google.com/patent/WO1999062546A1/en).

Hyaluronic acid can also be combined with other compounds such as chitosan and liposomes. These combinations generate molecules of high stability that increase high IgG titers when administered intranasally and intradermally [72]. Cationic liposome-hyaluronic acid hybrid nanoparticles were used in combination with *Yersinia pestis* antigens as an intranasal vaccine in mice. The results indicate that it was a more robust immune response, including a potent antibody response, revealing that hyaluronic acid, in combination with other compounds, could be used as an adjuvant for intranasal vaccines for respiratory illness [70]. 

## 9. Conclusions/Future Directions

While intravenous and intramuscular vaccinations stimulate systemic immunity, they fail to trigger mucosal immunity, which is crucial in a well-rounded and effective protective immunity for the individual, as well as to halt transmission of respiratory diseases. Intranasal mucosal delivery stimulates both of these responses, however, difficulties arise regarding vaccine retention in the nasal cavity, as well as the need for stimulation of the immune system, due to the fact that a more effective live attenuated vaccine would likely cause adverse side effects through the intranasal administrative route.

Biopolymers revealed great potential for their use as adjuvants in vaccine delivery systems, particularly for intranasal vaccines for respiratory diseases. Molecules such as starch, alginate, β-glucans, emulsan, hyaluronic acid and chitosan show promise as adjuvants for intranasal vaccines due to their effects of augmenting the immune response. Chitosan, gellan and alginate are advantageous due to their mucoadhesive properties, allowing better adhesion to the nasal cavity. Mucoadhesive antigens are able to increase the exposure time of the antigens, resulting in a robust immune response. Chitosan, emulsan, starch, hyaluronic acid and glucan particles containing β-glucan have the potential to be developed into antigen and adjuvant delivery systems that may help with all routes of administration, including intranasal.

Further research using these biopolymers, individually and combined with one another, has the potential to improve intranasal vaccines, and aid in decreasing the burden of disease due to communicable respiratory illness.

## Figures and Tables

**Figure 1 materials-12-02445-f001:**
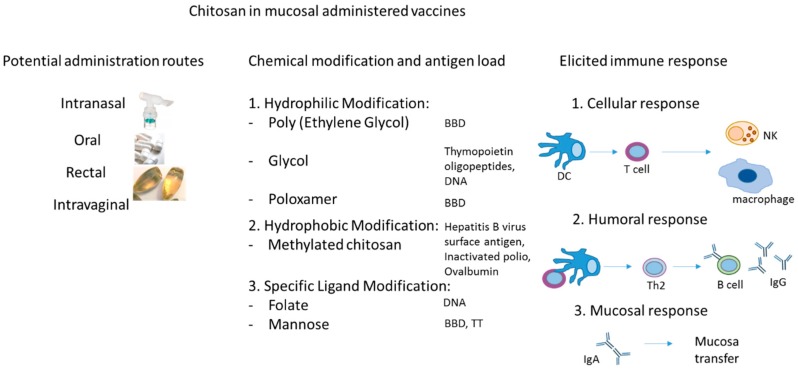
Properties of chitosan in the design of mucosal administered vaccines (i.e., administration mucosal routes, chemical modification for variate antigen load and applications, diverse elicited immune response). BBD = Bordetella bronchiseptica dermonecrotoxin, TT = Tetanus toxoid, DC = dendritic cell, Th2 = T helper 2 CD4 + cell, IgG = Immunoglobulin G antibodies, IgA = Immunoglobulin A antibodies.

**Table 1 materials-12-02445-t001:** Biopolymers with useful properties for vaccine design.

Biopolymer	Applications	Advantages for Vaccine Design	References
Chitosan	Hydrogels, dressings, drug delivery systems, gene delivery	Biocompatibility, degradability, mucoadhesive, mucosal absorption, stimulation of immune response, antimicrobial	[18,19,20]
Starch	Scaffolds, drug delivery systems, tissue repair	Biocompatibility, immune modulation, biodegradability	[18,21]
Alginate	Coatings and dressings, drug delivery, pharmaceutical excipient, packaging	Biocompatibility, immunomodulation, vaccine enhancer, tumor suppressor	[22,23]
Cellulose	Pharmaceutical excipient, packaging, drug delivery, coatings	Low price, abundant in nature, vaccine adjuvant, easy protein binding	[18,24]
Gellan	Food industry, multifunctional additive in pharmacy, regenerative medicine, gene delivery	Antigen delivery, mucosal delivery, immune response stimulation	[17,25]

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
