# Peer review of "Use of Biopolymers in Mucosally-Administered Vaccinations for Respiratory Disease"

_materials, 2019, doi:10.3390/ma12152445_

Reviewer 1 Report

The authors describe the application of some biopolymers as promising adjuvants for intranasal vaccines. The manuscript fits the scope of the journal. I do have a few remarks; appropriate addressing those would make the manuscript fit for publication.

Introduction

Line 43, has greatly improved the situation …

Gellan

Lines 144-153, it is not mentioned that the cited studies were (presumably) relating to mice.

Lien 151, see above comment; mice are mentioned here the first time in chapter "gellan".

Line 161, to add in a sentence or two: more data on different antigens is needed to be able to better estimate/predict the useability of this biopolymer in a larger scale (e.g. contradictory results as regards influenza A and B vaccination efficiency).

b-glucans/emulsan

Lines 184/205/207: potential

Conclusions/Future Directions

Line 238, Entities (or molecules) such as …

References

Please use journal abbreviations, indicate only volumes but not issues

Is there a reason to have two lists of references ((i), 1-35, (ii), i – xi (editorial remarks only)

Author Response

We would like to thank the reviewer for all the comments that have greatly improved our manuscript. Below are the detail answers to each of the comments but we would like to first express our gratitude for the careful consideration that this reviewer had while commenting on our manuscript

The authors describe the application of some biopolymers as promising adjuvants for intranasal vaccines. The manuscript fits the scope of the journal. I do have a few remarks; appropriate addressing those would make the manuscript fit for publication.

Introduction

Line 43, has greatly improved the situation …

Gellan

Answer: we have addressed this comment

Lines 144-153, it is not mentioned that the cited studies were (presumably) relating to mice.

Lien 151, see above comment; mice are mentioned here the first time in chapter "gellan".

Answer: we have added this information

Line 161, to add in a sentence or two: more data on different antigens is needed to be able to better estimate/predict the usability of this biopolymer in a larger scale (e.g. contradictory results as regards influenza A and B vaccination efficiency).

Answer: we have inserted additional comments and explanation 

b-glucans/emulsan

Lines 184/205/207: potential

Conclusions/Future Directions

Line 238, Entities (or molecules) such as …

References

Please use journal abbreviations, indicate only volumes but not issues

Is there a reason to have two lists of references ((i), 1-35, (ii), i – xi (editorial remarks only)

Answer: we have corrected the references and update them

Reviewer 2 Report

This manuscript presents a review of biopolymers for mucosally-delivered vaccines. The topic is very interesting and relevant, but the manuscript needs work to improve its quality. The review of the topic is largely superficial and does not get into much depth for the different biopolymers examined. The authors should elaborate on contents of Figure 1 and Table 1 to expand the information on the topic. The authors should also describe the mucosal regions of the nasal cavity in more depth, so an average reader can better appreciate the needs and challenges of vaccine delivery. Citing additional references would improve the quality of the manuscript. The references cited in Table 1 should be in the same reference list as the others and should be cited in the manuscript text.

Author Response

All the authors highly appreciate the comments of reviewer 2. We have expanded our manuscript and we hope that this is what reviewer 2 was expecting, we indeed feel that our paper has greatly improved with the addition of more detail information. Below is the answer to this reviewer comments.

This manuscript presents a review of biopolymers for mucosally-delivered vaccines. The topic is very interesting and relevant, but the manuscript needs work to improve its quality. The review of the topic is largely superficial and does not get into much depth for the different biopolymers examined. The authors should elaborate on contents of Figure 1 and Table 1 to expand the information on the topic. The authors should also describe the mucosal regions of the nasal cavity in more depth, so an average reader can better appreciate the needs and challenges of vaccine delivery. Citing additional references would improve the quality of the manuscript. The references cited in Table 1 should be in the same reference list as the others and should be cited in the manuscript text.

Answer: we added more information around the table and figure, and also tried to explain better some aspects related to vaccine delivery in respiratory formulations. References were also organized as requested.

Reviewer 3 Report

The manuscript is aimed to describe a plethora of biopolymers as vaccine adjuvants for mucosal administration against respiratory diseases.

Most of the ref. journals are not displayed in the correct abbreviation form.

References related to tables should be incorporated within the whole literature list for a more convenient approach towards readers.

It is strongly suggested to either modify the title or re-arrange the text: the core of the manuscript is focused on biopolymers. Discussion about their application for respiratory diseases is severely lacking. Overall, the review exhibits a potential interest, but it requires a more in-depth analysis and a more complete bibliography support.

At the present stage, the manuscript looks closer to a mini-review, rather than a complete review.

No TOC graphic is presented.

Author Response

We feel highly thankful for reviewer 3 suggestions. We added a figure that we hope will increase the impact of our manuscript and after we have read the revised version in which we have included all suggestions, we feel highly accomplished for the great improvement on the manuscript. Native English speakers have edited the manuscript and now it reads better. Equally, we had carefully revised our citations. We had also augmented the information included in the manuscript and we hope that the reviewer feels about the new version of the manuscript as happy as we are.

The manuscript is aimed to describe a plethora of biopolymers as vaccine adjuvants for mucosal administration against respiratory diseases.

Most of the ref. journals are not displayed in the correct abbreviation form.

 Answer: we have corrected and organize the references

References related to tables should be incorporated within the whole literature list for a more convenient approach towards readers.

 Answer: we have corrected

It is strongly suggested to either modify the title or re-arrange the text: the core of the manuscript is focused on biopolymers. Discussion about their application for respiratory diseases is severely lacking. Overall, the review exhibits a potential interest, but it requires a more in-depth analysis and more complete bibliography support.

At the present stage, the manuscript looks closer to a mini-review, rather than a complete review.

  Answer: we have expanded the reference list and provided additional explanations and info.

No TOC graphic is presented.

Answer: we have a potential graphical abstract bellow

Figure not ready for publication

Round  2

Reviewer 2 Report

The authors have addressed all of my concerns.

Reviewer 3 Report

The manuscript looks remarkably improved.

References have been re-arranged properly, but still the correct journal abbreviation for most of the cited articles is missing.

Nevertheless, an adequate English check shall be helpful, mainly over the newly added sentences: i.e., sentences should be shortened, appropriate punctuation would better allow the reader to more easily understand the delivered concepts.